# Tailored Exercise Intervention in Metabolic Syndrome: Cardiometabolic Improvements Beyond Weight Loss and Diet—A Prospective Observational Study

**DOI:** 10.3390/nu17050872

**Published:** 2025-02-28

**Authors:** Michele Braggio, Gianluigi Dorelli, Nicola Olivato, Vito Lamberti, Maria Teresa Valenti, Luca Dalle Carbonare, Mattia Cominacini

**Affiliations:** 1Department of Engineering for Innovation Medicine, University of Verona, 37100 Verona, Italy; michele.braggio@univr.it (M.B.); nicola.olivato@gmail.com (N.O.); luca.dallecarbonare@univr.it (L.D.C.); 2Department of Neurosciences, Biomedicine and Movement Sciences, University of Verona, 37100 Verona, Italy; gianluigi.dorelli@univr.it (G.D.); mariateresa.valenti@univr.it (M.T.V.); 3Sport Medicine and Motor Activity Institute c.FMSI-CONI, Vittorio Veneto, 31029 Treviso, Italy; vitorbras7@hotmail.it

**Keywords:** metabolic syndrome, personalized exercise, lipid metabolism, cardiovascular health, moderate-intensity training

## Abstract

**Background**: Metabolic syndrome (MS) is a cluster of cardiovascular and metabolic risk factors that increase the likelihood of both acute events and chronic conditions. While exercise has been shown to improve individual risk factors associated with MS; research on its effects on MS as an integrated condition remains limited. This study aims to evaluate the effectiveness of a 6-month Adapted Personalized Motor Activity (AMPA) program for improving the health outcomes of individuals with MS. **Methods**: Seventy-one sedentary participants with MS (mean age: 63 ± 9.4 years, 46.5% female) completed a 6-month intervention, incorporating moderate-intensity aerobic and resistance training. Each participant received a personalized exercise plan prescribed by a sports medicine physician. The training was monitored via telemetry to ensure safety. No dietary recommendations were provided during the intervention. Baseline and post-intervention assessments included Cardiopulmonary Exercise Testing (CPET), anthropometric measurements, blood pressure, heart rate, lipid profile (total cholesterol, HDL, LDL, and triglycerides), fasting glucose, and HbA1c. **Results**: Significant improvements were observed in fasting glucose (−10.6%, *p* < 0.001), HbA1c (−3.88%, *p* < 0.001), HDL cholesterol (+20.8%, *p* < 0.001), LDL cholesterol (−25.1%, *p* < 0.001), and VO_2_max (+8.6%, *p* < 0.001). Systolic and diastolic blood pressure also decreased significantly, with reductions of −12% (*p* < 0.001) and −5.9% (*p* < 0.001), respectively. Reductions in weight and waist circumference were statistically significant but modest and clinically irrelevant, showing no correlation with improvements in cardio-metabolic parameters. Logistic regression and correlation matrix analyses were performed to identify key predictors of changes in individual risk factors. **Conclusions**: While personalized exercise alone may not fully control individual risk factors of metabolic syndrome, its overall effect is comparable to low-intensity pharmacological polytherapy with minimal adverse effects. These benefits appear to be independent of dietary habits, gender, and both baseline and post-intervention physical performance and anthropometric measures.

## 1. Introduction

Metabolic syndrome (MS) refers to a combination of cardiovascular (CV) risk factors, including hyperglycemia, elevated blood pressure, dyslipidemia, and central obesity. Considering that cardiovascular diseases alone are responsible for nearly one-third of deaths globally [1], the prevention and management of MS is a significant public health issue, especially if we consider that the estimated global prevalence of MS ranges from 12.5% to 31.4% [2,3]. Studies reveal that regular exercise can deliver results similar to those achieved with blood pressure or blood sugar-lowering medications, benefiting both sedentary non-MS individuals and those with MS [4,5]. Exercise provides a wide range of benefits for subjects with and without MS. One of the most important factors is an individual’s ability to deliver and utilize oxygen during physical activity, known as maximal oxygen uptake (VO_2_max) [6]. Higher VO_2_max levels are strongly associated with lower overall mortality rates in healthy individuals and those with chronic conditions [7]. This means that improving VO_2_max through regular endurance exercise can enhance cardiovascular fitness, reduce the risk of disease progression, and ultimately increase longevity in healthy sedentary and MS populations [7]. Endurance training enhances body composition by increasing daily energy expenditure and decreasing visceral fat [4,6,8]. Meanwhile, resistance training helps prevent muscle loss during weight reduction while increasing fat-free mass, maximal strength, muscular efficiency, and energetics [6]. In addition to aiding weight loss, exercise enhances glucose metabolism by lowering blood sugar levels and reducing the risk of type 2 diabetes [9]. It increases skeletal muscle insulin sensitivity, improving the signaling for Glucose Transporter type 4 (GLUT4) translocation, and enhances mitochondrial protein expression [10]. While physical activity alone has a limited effect on cholesterol management, pairing it with lipid-lowering medication can maximize the outcomes [11]. Exercise primarily improves high-density lipoprotein cholesterol (HDL-C) levels; however, there is some discrepancy in the literature regarding its effect on reducing low-density lipoprotein cholesterol (LDL-C) [12,13]. Nonetheless, exercise can slow the progression of atherosclerosis by increasing the average size of atherogenic small dense LDL-C particles [14] and enhancing their resistance to oxidative processes, even in the absence of significant changes in serum LDL-C levels [15]. Regular physical activity also helps lower blood pressure, similar to common antihypertensive medications, especially for systolic blood pressure (SBP) [5]. This is obtained through two mechanisms: firstly, exercise reduces vascular resistance through an increased expression of endothelial nitric oxide synthase, a reduction in either nervous sympathetic activity and arterial stiffness; secondly, physical activity acts on the renin–angiotensin–aldosterone system, decreasing water retention [16,17]. There is a gap in the literature regarding whether exercise can improve all the mentioned parameters of metabolic syndrome (MS) despite no reduction in body weight, BMI, or waist circumference when following the American College of Sports Medicine (ACSM) guidelines [18]. While the overall effect of combined exercise on MS parameters is well-documented [19], no studies have thoroughly explored the relationship between the reduction in individual cardiovascular risk factors in MS. On the other hand, exercise training is often unobserved and its applicability on a large scale remains a major limitation. Among the various training programs proposed, Adapted Personalized Motor Activity (AMPA) is a structured intervention designed to improve VO_2_max, muscle mass, strength, and flexibility in patients with noncommunicable diseases. Notably, AMPA is one of the few programs that has been successfully implemented in large patient cohorts showing a good adherence rate, making it a valuable framework for further investigation [20,21]. However, despite being a cost-effective intervention for both preventing and mitigating the impact of metabolic syndrome (MS), exercise prescription remains underutilized. Given that exercise targets multiple disrupted mechanisms in MS patients, we hypothesize that its long-term effects may be more pronounced when combined with standard therapy. Furthermore, we propose that the impact of exercise on cardiovascular risk may occur among patients with higher baseline CV risk. Therefore, the present study aims to evaluate the extended effects of a six-month AMPA program on metabolic syndrome patients and quantify its overall efficacy independently from gender, dietary habits, and medications. Changing anthropometric, cardiopulmonary, and CV risk factors are considered.

## 2. Materials and Methods

### 2.1. Subjects

The study initially enrolled 230 subjects who met the inclusion criteria and had no contraindications to the physical activity outlined in the protocol. Following the 6-month AMPA program, the study population was defined by selecting individuals diagnosed with metabolic syndrome according to IDF criteria [22,23]. The study was conducted in accordance with the Declaration of Helsinki and approval was obtained from the relevant ethics committee (protocol n° 1538, version 3). All participants were of Caucasian ethnicity and provided informed consent after receiving detailed information about the potential risks, benefits, and procedures involved.

### 2.2. Inclusion Criteria

The inclusion criteria included those aged between 40 and 75, diagnosis of metabolic syndrome according to the National Cholesterol Education Program Adult Treatment Panel III [24] criteria, and no participation in structured physical activity programs within the six months prior to the study.

### 2.3. Exclusion Criteria

The exclusion criteria included those with a history of musculoskeletal, neurological, or orthopedic disorders in the preceding six months that could hinder participation in the experimental protocol, acute cardiovascular conditions contraindicating physical activity, active cancer, infectious diseases, chronic obstructive pulmonary disease or active smoking; inability to provide informed consent.

### 2.4. Experimental Design

This study utilized a prospective observational design, aiming to assess the effects of the AMPA program on various health outcomes. The primary objectives were to evaluate the impact of the exercise program on metabolic parameters (fasting glycemia, HbA1c, LDL cholesterol, HDL cholesterol, and triglycerides), anthropometric measures (weight, BMI, waist, and hip circumference), and cardiopulmonary performance (HR, blood pressure, FVC, FEV1, and VO_2_). These outcomes were evaluated both at the baseline and after 6 months of participation in the exercise program. Participants underwent a baseline medical screening that included the collection of medical history, a complete clinical evaluation, anthropometric measurements, blood tests, and cardiopulmonary assessments. Anthropometric measurements included body weight (kg), height (cm), and waist circumference (cm). Body weight and height were measured in light clothing without shoes using a calibrated scale and stadiometer, respectively, to the nearest 0.1 kg and 0.5 cm. The Body Mass Index (BMI, kg/m^2^) was calculated. Cardiopulmonary fitness was assessed using a maximal graded exercise test on a cycle ergometer [25]. VO_2_max (mL/kg/min) was measured to determine cardiorespiratory fitness. Heart rate, blood pressure, and oxygen saturation were monitored throughout the test.

### 2.5. Training Protocol

The experimental group participated in a six-month tailored exercise program, that combined aerobic and resistance exercises. The program followed the guidelines of the American College of Sports Medicine (ACSM), with sessions supervised by exercise specialists to ensure safety, adherence, and proper progression [6].

#### 2.5.1. Exercise Monitoring and Adherence

The exercise program was conducted using various cardio and isotonic machines, including a Treadmill, Elliptical, Upright bike, Recumbent bike, as well as compressed-air isotonic machines such as Ercolina, Pectoral Machine, Lower-back, Leg-Press, Adductor Machine, Deltoids Press, and Lat Pull Down Convergent. These machines ensure proper progression of work and adherence to the necessary time intervals, with the ability to adjust the load to as low as 2 kg. All of the Panatta Sport Air Machines were equipped with an advanced IT system (Net Tutor Pro software 2013) and connected to a video terminal. This system allowed for the customization of training plans, real-time monitoring of the exercise data (e.g., heart rate, work duration), and real-time modification of the program based on the patient’s progress. Additionally, the system created a digital archive of the sessions. To ensure consistent adherence to the individualized training plan, each patient was provided with a personal pen drive (touch probe) that stored their specific exercise program. Once connected to the machine’s interface, the pen drive activated the appropriate settings for that session. Furthermore, the machines were equipped with a heart rate monitoring system that could automatically stop the exercise if the maximum recommended heart rate was exceeded. In case of an emergency, the machines could be linked to diagnostic tools, including ECG, spirometer, and cardiac ultrasound. The medical gym was also equipped with a semi-automatic external defibrillator (AED) and a medical trolley for emergency situations.

#### 2.5.2. Aerobic Training

Conducted three times per week, focusing on moderate-intensity exercises aimed at 55–70% of VO_2_max or 60–80% of the participant’s maximum heart rate. Aerobic activities included treadmill walking, cycling, and elliptical training. Each session lasted 25–30 min, gradually increasing in intensity as the participants’ fitness improved [6,26].

#### 2.5.3. Resistance Training

Conducted twice per week, focusing on strengthening the major muscle groups. The resistance training regimen included 2–3 sets of 8–15 repetitions for each major muscle group, with 1–3 min of rest between sets. The resistance load was set at 50–70% of the participant’s one-repetition maximum (1-RM) [6,27]. Weight machines were used to ensure correct form and safety during the exercises. Each session included a warm-up: 5–10 min of low-intensity aerobic exercise to prepare the body for the workout [6], and a cool-down: 5–10 min recovery phase, consisting of light aerobic exercise and stretching [6].

### 2.6. Statistical Analysis

Continuous variables are presented as mean ± SD or median with range, depending on the distribution (assessed by the Shapiro–Wilk test). Comparisons between groups for normally distributed data were performed using Student’s *t*-test or paired *t*-test, as appropriate. Welch’s *t*-test was applied for variables with unequal variances. Nominal variables are presented as counts and percentages, with group comparisons made using the chi-square test or Fisher’s exact test. Correlations between continuous variables were assessed using Pearson or Spearman correlation coefficients. Linear and logistic regression models were applied as appropriate. A *p*-value of <0.05 was considered statistically significant. Analysis was performed using Jamovi software version 2.3.19.0 [28,29].

## 3. Results

In the AMPA project, 230 sedentary participants of both sexes (118 men and 112 women) were enrolled, of whom 203 met the inclusion criteria beyond the diagnosis of metabolic syndrome and were deemed eligible for sports therapy, while 27 were excluded due to clinically unstable chronic diseases considered high risk or absolute contraindications. A total of 178 subjects (87.7%) completed the 6-month exercise program. Among these, 71 participants met at least three of the five criteria for metabolic syndrome diagnosis, as defined by the International Diabetes Federation (IDF). The cohort had a mean age of 63 ± 9.4 years, with 33 participants (46.5%) being female. The prevalence of individual risk factors was as follows: arterial hypertension in 63 participants (88.7%), hypertriglyceridemia in 25 (35.2%), low HDL cholesterol in 51 (71.8%), visceral obesity in 58 (81.2%), and impaired fasting glucose in 61 (85.9%). Additionally, 39 participants (54.9%) were classified as obese (BMI > 30 kg/m^2^), and 23 (32.4%) had type 2 diabetes (Figure 1). Regarding the distribution of metabolic syndrome diagnostic criteria, 34 participants (47.9%) met three criteria, 25 (35.2%) met four criteria, and 12 (16.9%) met all five criteria. Moreover, exercise-induced hypertension was observed in 28 subjects (39.4%) during the cardiopulmonary exercise test.

A significant improvement was observed in most of the parameters pre- and post-intervention, including anthropometric characteristics, glucose and lipid metabolism, and cardiorespiratory performance (Figure 2). However, the most notable effect sizes were observed in the following: fasting glucose (−10.6%, *p* < 0.001), total cholesterol (−10%, *p* < 0.001), HDL cholesterol (+20.8%, *p* < 0.001), LDL cholesterol (−25.1%, *p* < 0.001), forced vital capacity and forced expiratory volume in 1 s (both +6.6%, *p* < 0.001), VO_2_max (+8.6%, *p* < 0.001), indexed VO_2_max (+8.8%, *p* < 0.001), aerobic threshold (+10.4%, *p* < 0.001), maximal oxygen pulse (+7.6%, *p* < 0.001), resting heart rate (−7.6%, *p* < 0.001), and recovery phase heart rate (−6%, *p* < 0.001), as well as resting systolic and diastolic blood pressure (−12% and −5.9%, respectively, both *p* < 0.001), recovery phase systolic and diastolic blood pressure (−8% and −6.7%, respectively, both *p* < 0.001), and diastolic blood pressure at peak exertion (−5.9%, *p* < 0.001). No significant difference was observed in heart rate or systolic blood pressure at peak exertion, suggesting that the cardiopulmonary tests were maximal both at baseline and after the intervention. Significant but clinically negligible changes were observed in the number of pills taken, anthropometric characteristics, ventilatory reserve, Tiffeneau index, relative peak power output, and age-adjusted peak power output.

The clinical characteristics, the results of the biochemical and instrumental tests at baseline and after 6 months of AMPA, as well as the absolute and percentage changes in these variables between pre- and post-intervention, are shown in Table 1.

### 3.1. Correlations

A significant positive correlation was observed between the percentage changes in triglycerides and baseline HDL cholesterol (r = 0.454, *p* < 0.001), suggesting that participants with lower baseline HDL cholesterol experienced greater reductions in triglyceride levels. Conversely, a significantly negative correlation was identified between age and percentage changes in diastolic blood pressure (r = −0.348, *p* = 0.003), indicating that older participants showed more substantial improvements in this parameter.

### 3.2. Gender Differences

Significant baseline differences were found, with men taking a higher number of pills (ES = 0.66, *p* = 0.007), having higher fasting glucose levels (ES = 0.50, *p* = 0.042), lower HDL cholesterol (ES = 0.633, *p* = 0.010), higher triglycerides (ES = 0.62, *p* = 0.039), and a lower resting heart rate (ES = 0.60, *p* = 0.014) compared to women. In men, beyond higher fasting glucose levels, a history of type 2 diabetes was also more common (44.7% in men vs. 21.2% in women; χ^2^ = 4.37, *p* = 0.037). Conversely, a higher incidence of exercise-induced hypertension was observed in women (26.3% in men vs. 54.5% in women; χ^2^ = 5.89, *p* = 0.015). Beyond the notable baseline anthropometric, laboratory and cardiopulmonary fitness differences, no significant gender differences were observed in the percentage changes in most anthropometric measures, blood exams, and cardiopulmonary fitness. However, men showed greater reductions in triglycerides (ES = 0.69, *p* = 0.022) and experienced less reduction in systolic blood pressure at peak exercise (ES = 0.7, *p* = 0.004) and during recovery (ES = 0.57, *p* = 0.019).

### 3.3. Linear Regression Models

Regression models evaluated the relationship between baseline values of risk factors and their percentage changes, with separate models constructed for each percentage change. Each model was adjusted for baseline values, including the baseline of the variable undergoing change, and included age and gender as covariates (Table 2). Notably, gender did not significantly influence any of the percentage changes in the variables studied.

### 3.4. Key Findings from the Models

The percentage change in LDL was significantly associated with its baseline level (β = −0.377, SE = 0.110, *p* = 0.001), indicating that higher baseline LDL levels were associated with greater reductions. Changes in triglycerides showed significant associations with both baseline triglyceride levels (β = −0.233, SE = 0.081, *p* = 0.007) and fasting glucose (β = −0.411, SE = 0.129, *p* = 0.003). Higher baseline values of these variables were linked to greater reductions in triglycerides. The percentage change in HDL was significantly associated with age (β = −0.655, SE = 0.252, *p* = 0.013) and baseline HDL levels (β = −0.847, SE = 0.279, *p* = 0.004). Older age and higher baseline HDL levels were associated with smaller percentage increases in HDL. These results highlight that baseline levels and age are key determinants of changes in these risk factors, while gender showed no significant influence.

## 4. Discussion

In this study, we evaluated the effects of adapted and monitored exercise in 71 MS patients, following ACSM’s guidelines [6]. Our results show that Adapted Personalized Motor Activity (AMPA) [21] positively impacts MS and cardiovascular risk factors. Adherence to the program was high (77.4%), with less than one-fourth of participants unable to complete the six-month exercise program. Other studies reported higher dropout rates [30,31]. We attribute the low dropout rate to a one-month familiarization period with low-intensity exercises included in the assisted AMPA program, and a variety of gym equipment. The program comprised resistance and aerobic exercises three times a week for 50–60 min per session. Moderate-intensity exercise (150 min weekly) is known to improve MS, with better results at higher intensity/volume levels [32]. After the familiarization with AMPA, aerobic intensity was initially moderate, progressing to 75% of VO_2_max, while resistance training intensity ranged from 50 to 70% of maximum voluntary muscle contraction. The recent CardioRACE trial [33] found that heart rate reserve (HRR)-based combined aerobic and resistance training reduced cardiovascular disease risk but did not significantly affect SBP or LDL cholesterol. In contrast, our CPET-based and personalized training in MS patients showed superior outcomes compared to HRR-based or standardized training [30]. The most significant effect of AMPA was on lipid profiles, particularly a reduction in LDL levels (−27.3 mg/dL). Higher baseline LDL was associated with greater reductions, with new perspectives on emphasizing the benefits of physical activity in MS. A recent meta-analysis reported a −7.22 mg/dL LDL reduction with exercise training [13], though it excluded patients with CV diseases and found moderate-to-high heterogeneity among studies. Interestingly, older participants exhibited lower increases in HDL, showing an inverse correlation between age and HDL levels, confirming that exercise may have a reduced impact on lipid profiles in the elderly [34]. However, participants with low baseline HDL experienced better lipid improvements, particularly in triglyceride reduction among men, highlighting the importance of exercise for MS patients with poor lipid profiles. On the other hand, physical activity is known to reduce visceral adipose tissue (VAT) and influence lipid metabolism (i.e., lipoprotein lipase, PCSK) [35] and inflammation [36]. While our patients showed significant reductions in weight and waist circumference, these changes did not correlate with cardio-metabolic improvements. Our study corroborates [10] that exercise positively influences body composition and metabolism even without anthropometric changes. Since the aim of this study was to isolate the effects of exercise, no dietary modifications were implemented, as diet plays a critical role in reducing MS components [37]. Cardiovascular fitness improved significantly, with increases in VO_2_max (+8.6%), AT (+10.4%), and peak O_2_ pulse (+8.8%). VO_2_max is an independent CV risk factor in MS [38,39], correlating with lipid profiles [40]. CPET confirmed normal ventilatory reserve before and after AMPA, and training improved resting pulmonary function parameters FVC and FEV1 [41]. No ventilatory limitations were detected. Greater fitness improvements were observed with AMPA compared to non-monitored exercise programs [31] or HRR-based aerobic exercise [30]. Sedentary lifestyles contribute to non-transmittable chronic disease development, with exercise recognized as medicine for these conditions [42]. Cardiorespiratory fitness in MS does not improve by merely reducing sedentary behavior [43]. Medication use remained unchanged after AMPA, with exercise effects comparable to low-intensity pharmacological polytherapy [44]. The program also reduced serum uric acid, associated with insulin resistance, hypertension [45], and it is related cardiomyopathy [46]. Fasting blood glucose and HbA1c also improved (−10.6% and −3.88%), with greater effects in those with higher baseline values. It is known that aerobic exercise increases glucose uptake in skeletal muscle [47], while resistance training enhances insulin sensitivity by increasing muscle mass [48]. No significant gender differences were observed in anthropometric, blood, or fitness changes, likely due to our mainly postmenopausal female cohort [49]. However, men exhibited greater triglyceride reductions and higher fasting glucose levels correlated with greater triglyceride decreases. In fact, reduced insulin resistance is related to improvements in skeletal muscle fatty acid oxidation [50]. These data support that physical activity can act on different metabolic pathways with pleiotropic effects in MS [51] as well as on cardiorespiratory fitness parameters (i.e., VO_2_max) [52]. The latter aspect is of particular interest in MS because a higher VO_2_max may protect from hypertension development [53]. SBP and DBP decreased significantly (−12% and −5.9%), with older participants showing greater DBP reductions, possibly due to lower initial fitness and arterial stiffness modulation [53]. Data on the duration of intervention to gain effectiveness are conflicting (3, 6, 12 months) [54,55] but in the specific setting of MS as in our cohort, 6 months seems to have a significant reduction in blood pressure in the elderly [56]. Exercise-induced hypertension (EIH) was more common in women, yet AMPA led to a reduction in peak SBP, confirming its safety and effectiveness [57]. Remarkably, EIH has been correlated with sudden cardiac death, CV accidents [58], and pathological vascular stiffness in middle-aged women [59]. AMPA likely improved blood pressure via nitric oxide modulation [59] and insulin sensitivity [60]. Similarly, the effects of lower sympathetic tone positively influenced resting heart rate (−7.6%), recovery heart rate (−6%), and blood pressure [60]. Therefore, our results indicate that Adapted Personalized Motor Activity is comparable to low-intensity pharmacological polytherapy in reducing blood glucose [61], while also improving lipid profiles [62] and lowering blood pressure [63]. This is particularly significant, as these effects are known to contribute to reduced mortality and overall health improvements through metabolomic pathways, as observed in various animal models [64]. Based on our findings, we assert that exercise should be promoted as an adjunct to standard therapy in MS patients, as they experience different exercise-related metabolomic modifications compared to healthy individuals [65]. Future research should integrate clinical findings with genetic predisposition and epigenetic responses to exercise to further unravel the complexity of MS.

This study has some limitations. First, this study lacked a baseline and post-exercise body composition measurements, which could play a crucial role in reducing CV risk. Additionally, we did not include a control group, as the study aimed to evaluate prospectively CV risk modifications in patients with MS, focusing on the synergistic effect of combined exercise and standard therapy. Other variables, such as changes in dietary habits, may have influenced our results. However, the monitored training program was conducted by healthcare personnel, which likely minimized intra-cohort variations. Given the high prevalence of MS, a sample size of 71 patients may be considered small. Additionally, a highly monitored program like AMPA may not always be feasible in routine clinical practice. Future research should focus on larger intervention groups to enhance generalizability and applicability. Lastly, sedentary habits are known to contribute to MS development, meaning our findings may not be generalizable to individuals with different baseline fitness levels. However, this highlights the importance of tailored exercise programs for MS patients.

## 5. Conclusions

Our study demonstrates that improvements in CV risk factors among patients with MS, including lipid profiles, fasting blood glucose, and blood pressure, are not significantly influenced by gender or weight loss, even in the absence of dietary intervention. However, individuals with higher CV risk experience greater benefits from combined, tailored, and supervised exercise training. Our findings highlight a significant reduction in CV risk markers among MS adults, suggesting that the overall effect of Adapted Personalized Motor Activity (AMPA) is comparable to that of low-intensity pharmacological polytherapy. Future research should focus on personalized exercise prescriptions for MS, employing a long-term, multi-omics approach to better understand the complexity of this condition and the potential variability in therapeutic exercise responses. Based on our results, we strongly advocate for the integration of CPET-based, tailored exercise prescriptions in clinical practice, especially for MS patients at high CV risk.

## Figures and Tables

**Figure 1 nutrients-17-00872-f001:**
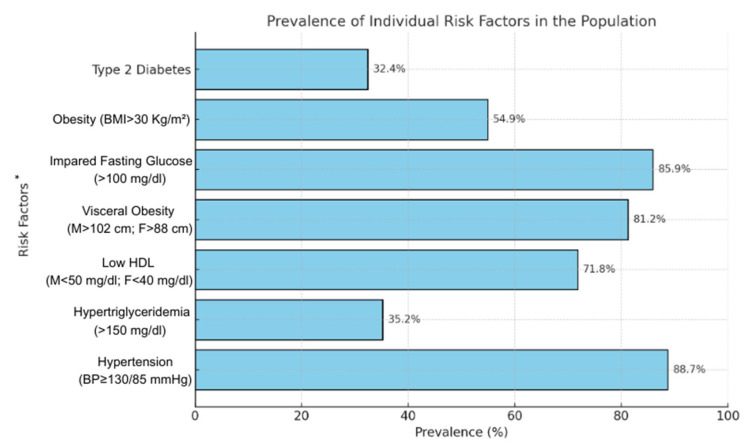
Prevalence of individual metabolic syndrome risk factors in the study population. BMI, body mass index; HDL, high density lipoprotein; * A risk factor is considered present even when the patient is undergoing specific treatment for that condition.

**Figure 2 nutrients-17-00872-f002:**
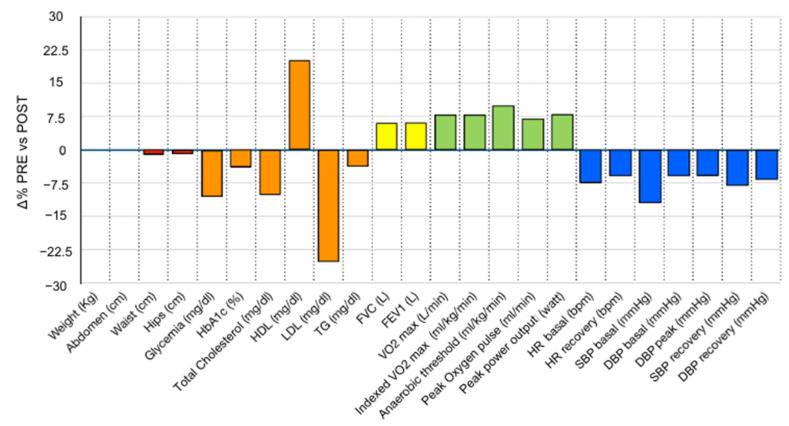
Percentage changes in the main variables observed after the AMPA intervention. HbA1c%, glycated hemoglobin; HDL, high density lipoprotein; LDL, low density lipoprotein; TG, triglycerides; FVC, forced vital capacity; FEV1, forced expiratory volume in 1 s; VO_2_max, maximum oxygen consumption rate; Indexed VO_2_max, maximal oxygen consumption rate normalized to body weight; HR, heart rate; SBP, systolic blood pressure; DBP, diastolic blood pressure.

**Table 1 nutrients-17-00872-t001:** Clinical characteristics, laboratory tests, and instrumental assessments at baseline and after 6 months of the AMPA program.

	Pre Intervention Mean (±SD) or Median (Min–Max)	Post Intervention Mean (±SD) or Median (Min–Max)	*p*	ΔPre-Post Mean (±SD) or Median (Min–Max)	Δ%Pre-Post Mean (±SD) or Median (Min–Max)
N° of pills	3.0 (0–15)	3.0 (0–15)	0.025	0 (−3–2)	-
Weight (kg)	84.0 (47–125)	84.0 (48–125)	0.03	0 (−11–5)	0 (−15.3–4.8)
Waist (cm)	103.0 (65–131)	102.0 (68–132)	<0.001	−1 (−12–6)	−1.1 (−13.5–8.5)
Hips (cm)	106.0 (74–142)	104.0 (85–142)	0.008	−1 (−14–11)	−1 (−14.6–12.9)
Glycemia (mg/dL)	120.0 (86–340)	107.0 (78–246)	<0.001	−11 (−144–65)	−10.6 (−73.5–34.9)
HbA1c (%)	6.6 (2.83–14.6)	6.4 (4.9–9.7)	<0.001	−0.2 (−6.2–1.8)	−3.88 (−73.8–18.6)
Total Cholesterol (mg/dL)	205.0 (79–324)	182.0 (115–278)	<0.001	−19 (−87–98)	−10 (−54–55)
HDL (mg/dL)	41.5 (±11.3)	53.9 (±13.4)	<0.001	12.47 (±11.64)	20.8 (±18.6)
LDL (mg/dL)	148.9 (±34.8)	121.6 (±25.8)	<0.001	−27.3 (±33.3)	−25.1 (±30.3)
TG (mg/dL)	152.0 (46–403)	141.0 (58–459)	0.18	−5 (−206–89)	−3.6 (−191.7–52.5)
Uric acid (mg/dL)	5.9 (±1.3)	5.6 (±1.1)	0.012	-	-
Creatinine (mg/dL)	0.83 (0.53–1.55)	0.87 (0.62–1.74)	0.48	-	-
FVC (L)	3.27 (2.04–5.95)	3.51 (1.77–6.07)	<0.001	0.2 (−1.59–1.73)	6.6 (−50–39.7)
FVC %	104.5 (56–196)	111.7 (62–157)	<0.001	-	-
FEV1 (L)	2.50 (1.1–4.4)	2.68 (1.35–4.66)	<0.001	0.18 (−1.65–1.87)	6.6 (−64.7–42.6)
FEV1 %	95.5 (62–205)	106.5 (62–154)	<0.001	-	-
Tiffeneau Index	0.78 (0.48–0.91)	0.78 (0.65–0.95)	0.083	-	-
PEF (L)	6.32 (2.72–13.54)	6.42 (3.01–11.17)	0.74	-	-
PEF%	93.5 (55–141)	95.6 (49.2–134)	0.7	-	-
VO_2_max (L/min)	1.42 (0.75–2.76)	1.51 (0.81–2.90)	<0.001	0.150 (−0.660–0.790)	8.6 (−31.4–33.8)
iVO_2_max (mL/kg/min)	16.8 (10.8–27.6)	18.6 (12.3–33.9)	<0.001	1.7 (−6.6–11.1)	8.8 (−39.8–32.8)
AT (mL/kg/min)	12.83 (±2.45)	14.28 (±2.75)	<0.001	1.61 (±1.92)	10.4 (−47.5–39)
Peak O_2_ pulse (mL/min)	12.0 (6–22)	13.0 (7–23)	<0.001	1 (−8–12.5)	7.6 (−57.1–54.4)
Ventilatory reserve (%)	52.91 (±12.70)	50.86 (±11.78)	0.19	−2.06 (±13.24)	−2.06 (±5.4)
Age-adjusted PPO (watt)	137.0 (63–262)	134.0 (65–262)	0.01	0 (−48–10)	0 (−27.1–5.5)
PPO (watt)	110.0 (60–224)	125.0 (75–250)	<0.001	12 (−29–56)	8.3 (−19.3–32.7)
Relative PPO (watt/kg)	1.46 (±0.39)	1.61 (±0.39)	<0.001	0.16 (± 0.32–0.61)	0 (−27.1–5.47)
HR basal (bpm)	68.0 (48–98)	61.0 (46–83)	<0.001	−5 (−29–15)	−7.6 (−49.2–22.4)
HR peak (bpm)	121.5 (±17.2)	122.1 (±16.5)	0.7	-	-
HR recovery (bpm)	89.4 (±15.9)	83.8 (±11.3)	<0.001	−5.6 (± 10.1)	−6 (−40.9–20.3)
SBP basal (mmHg)	140.0 (110–160)	120.0 (105–160)	<0.001	−15 (−55–30)	−12 (−52.4–20)
DBP basal (mmHg)	80.0 (70–110)	80.0 (60–95)	<0.001	−5 (−35–15)	−5.9 (−50–17.7)
SBP peak (mmHg)	195.0 (150–250)	190.0 (155–240)	0.2	-	-
DBP peak (mmHg)	85.0 (55–110)	80.0 (60–110)	0.004	−5 (−30–30)	−5.9 (−38.5–27.8)
SBP recovery (mmHg)	139.6 (±13.8)	129.3 (±12.0)	<0.001	−10 (−40–25)	−8 (−30.4–17.2)
DBP recovery (mmHg)	80.0 (55–105)	75.0 (50–95)	<0.001	−5 (−25–15)	−6.7 (−41.7–17.7)

HbA1c%, glycated hemoglobin; HDL, high density lipoprotein; LDL, low density lipoprotein; TG, triglycerides; FVC, forced vital capacity; FEV1, forced expiratory volume in 1 s; PEF, peak expiratory flow; VO_2_max, maximum oxygen consumption rate; iVO_2_max, indexed maximal oxygen consumption rate; AT, aerobic threshold; PPO, peak power output; HR, heart rate; SBP, systolic blood pressure; DBP, diastolic blood pressure.

**Table 2 nutrients-17-00872-t002:** Summary of linear regression models for percentage changes in risk factors.

Outcome	Baseline Predictors	β	SE	*p*-Value
LDL cholesterol (%Δ)	Baseline LDL	−0.377	0.11	0.001
Triglycerides (%Δ)	Baseline triglycerides	−0.233	0.081	0.007
	Baseline fasting glucose	−0.411	0.129	0.003
HDL cholesterol (%Δ)	Age	−0.655	0.252	0.013
	Baseline HDL	−0.847	0.279	0.004
Fasting glucose (%Δ)	Baseline fasting glucose	−0.295	0.061	<0.001
Systolic blood pressure (%Δ)	Baseline LDL	0.089	0.045	0.057
Diastolic blood pressure (%Δ)	Baseline triglycerides	0.037	0.02	0.077

HDL, high density lipoprotein; LDL, low density lipoprotein.

## Data Availability

The data presented in this study are available upon request from the corresponding author due to privacy restrictions.

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
