# Peer review of "Tailored Exercise Intervention in Metabolic Syndrome: Cardiometabolic Improvements Beyond Weight Loss and Diet—A Prospective Observational Study"

_nutrients, 2025, doi:10.3390/nu17050872_

Round 1

Reviewer 1 Report

Comments and Suggestions for Authors

The introduction highlights the benefits of exercise for MS but does not clearly define gaps or controversies in existing research, please highlight the gap in existing research

I notice we don't have group control. Why? The lack of a control group makes it difficult to determine whether the observed changes are due to the intervention or other lifestyle factors.

The choice of 55-70% VOâ‚‚ max for aerobic training and 50-70% of 1-RM for resistance training is not justified. Why not higher or lower intensities?

The study claims to focus solely on exercise, but without tracking dietary habits, it is difficult to rule out dietary changes as a confounder. Did you consider tracking dietary habits to rule out their influence on the results?

The authors claim that exercise effects are “comparable to low-intensity pharmacological polytherapy,” yet no direct comparison to pharmacological interventions was performed. This statement should be softened or supported by literature.

The conclusion succinctly summarizes the findings but lacks a clear take-home message regarding clinical applicability. Additionally, it does not adequately address the limitations of the study, particularly the lack of dietary control and body composition analysis. The authors suggest "long-term, multi-omics approaches," but without clarifying what specific biological pathways should be studied.

Author Response

The introduction highlights the benefits of exercise for MS but does not clearly define gaps or controversies in existing research, please highlight the gap in existing research.

We appreciate the reviewer’s insightful comment. In response, we have revised the Introduction to explicitly highlight the gaps in existing research regarding the independent effects of structured exercise interventions on metabolic syndrome, particularly in the absence of dietary interventions and pharmacological treatments. We have also included relevant references to support this discussion.

I notice we don't have group control. Why? The lack of a control group makes it difficult to determine whether the observed changes are due to the intervention or other lifestyle factors.

The absence of a control group is due to the original design of the AMPA program, which was developed to provide a supervised exercise intervention from a secondary and tertiary prevention perspective. Given that the beneficial effects of exercise are well established across various clinical settings, the focus was placed on optimizing patient compliance rather than on comparing with a non-exercising control group. Moreover, the use of a control group without an active intervention aimed at managing risk factors would be ethically challenging in a clinical setting such as metabolic syndrome and might not have been approved by an ethics committee. We acknowledge this as a limitation of our study and have included a discussion on this aspect: 'We did not include a control group, as the study aimed to evaluate prospectively CV risk modifications in patients with MS, focusing on the synergistic effect of combined exercise and standard therapy.

The choice of 55-70% VOâ‚‚ max for aerobic training and 50-70% of 1-RM for resistance training is not justified. Why not higher or lower intensities?

The choice of 55–70% VOâ‚‚ max for aerobic training and 50–70% of 1-RM for resistance training was based on the ACSM Guidelines (11th edition), as cited multiple times in the manuscript. The ACSM recommendations support these intensity levels to maximize the benefits of an exercise program while ensuring safety and adherence in populations with metabolic syndrome. Our objective was not to test a novel exercise regimen but to apply an evidence-based, guideline-driven program to our specific cohort.

As with all guidelines, there may be limitations and the need for individualized interpretation. In this regard, cardiopulmonary exercise testing and field tests were used as effective tools to tailor exercise intensity, aligning with the ACSM guidelines: "To reduce the impact of MetS, variables that are considered risk factors for CVD and DM, initial exercise training should be performed at a moderate intensity (i.e., 40%–59% V∙ O2R or HRR) totaling a minimum of 150 min ∙ wk−1 or 30 min ∙ d−1 most days of the week to allow for optimal health/fitness improvements. When appropriate, progress to a more vigorous intensity (i.e., ≥60% V∙ O2R or HRR)." These recommendations provided the rationale for our selected intensity ranges, ensuring both efficacy and safety for our study population.

The study claims to focus solely on exercise, but without tracking dietary habits, it is difficult to rule out dietary changes as a confounder. Did you consider tracking dietary habits to rule out their influence on the results?

We acknowledge the potential influence of dietary habits on metabolic and cardiovascular outcomes. However, our study was specifically designed to assess the effects of a structured exercise intervention, independent of dietary modifications. While we did not track detailed dietary intake, it is important to note that participants were instructed to maintain their habitual diet throughout the study period. Moreover, previous evidence suggests that exercise alone can provide significant metabolic and cardiovascular benefits, even in the absence of dietary changes. We recognize this as a limitation and have included it in the discussion.

The authors claim that exercise effects are “comparable to low-intensity pharmacological polytherapy,” yet no direct comparison to pharmacological interventions was performed. This statement should be softened or supported by literature.

We appreciate the reviewer’s comment. To address this concern, we have added references to the literature supporting the statement that structured exercise programs can yield benefits comparable to those of low-intensity pharmacological polytherapy in managing metabolic syndrome and cardiovascular risk factors. These references provide evidence of exercise-induced improvements in glycemic control, lipid profile, and blood pressure regulation, similar to those achieved with pharmacological approaches. We have also revised the statement to ensure clarity and accuracy in the discussion.

The conclusion succinctly summarizes the findings but lacks a clear take-home message regarding clinical applicability. Additionally, it does not adequately address the limitations of the study, particularly the lack of dietary control and body composition analysis. The authors suggest "long-term, multi-omics approaches," but without clarifying what specific biological pathways should be studied.

We appreciate the reviewer’s suggestion and have revised the conclusion to enhance clarity regarding the clinical applicability of our findings. Specifically, we have emphasized how tailored exercise interventions can serve as an effective strategy for managing cardiovascular risk factors in metabolic syndrome, even in the absence of significant weight loss. Additionally, we have explicitly addressed the study’s limitations, including the lack of dietary control and body composition analysis, acknowledging their potential influence on the results. Regarding the suggestion for long-term multi-omics approaches, we have clarified which specific biological pathways warrant further investigation, particularly those related to metabolic adaptations, inflammation, and cardiovascular remodeling induced by exercise.

Reviewer 2 Report

Comments and Suggestions for Authors

1
Title The title of your manuscript should be concise, specific, and relevant. It should identify if the study reports (human or animal) trial data or is a systematic review, meta-analysis, or replication study. https://www.mdpi.com/journal/nutrients/instructions
Abstract The Abstract provides a sentence to place the question addressed in a broader context and highlight the purpose of the study. The Abstract explicitly states the study's main objective: to evaluate the effectiveness of a 6-month Adapted Personalized Motor Activity program in improving health outcomes in individuals with metabolic syndrome. A full stop is missing at the end of a sentence in line 35. Please write „The training” instead of „Training” in line 21.
1. Introduction The Introduction effectively describes the rationale for the work in the context of existing knowledge. The Introduction provides an explicit statement of the objective by stating that the present study aims to evaluate the extended effects of a six-month Adapted Personalized Motor Activity (AMPA) program on patients with metabolic syndrome. It also specifies the goal of quantifying the program's overall efficacy independently of gender, dietary habits, and medications. The Introduction effectively defines metabolic syndrome and its key components, establishing its relevance as a public health issue. Citing global prevalence rates and the association of cardiovascular diseases with mortality is good. However, more specific studies might be needed to improve the reliability of the effectiveness of exercise compared to medications. One of the keywords is „personalized exercise". This term does not exist in the Introduction. Please delete „the” before „management” in line 43. Insert a comma before „and central obesity” in line 41. Lines 51-54: The following does not read well: „This means that improving VO2max through regular endurance exercise can play a critical role in enhancing cardiovascular fitness, reducing the risk of disease progression, and ultimately increasing longevity in both healthy sedentary and MS populations [7].” Maybe the Authors want to use the following instead: „This means that improving VO2max through regular endurance exercise can enhance cardiovascular fitness, reduce the risk of disease progression, and ultimately increase longevity in healthy sedentary and MS populations [7].”
2
The following does not read well: „Endurance training helps to improve body composition by boosting daily energy expenditure and reducing visceral fat [4,6,8]; resistance training prevents muscle loss during weight reduction, increase fat-free mass, as well as maximal strength, muscular efficiency and energetics [6].” Lines 54-57. Maybe the Authors want to use the following instead: „Endurance training enhances body composition by increasing daily energy expenditure and decreasing visceral fat [4,6,8]. Meanwhile, resistance training helps prevent muscle loss during weight reduction while increasing fat-free mass, maximal strength, muscular efficiency, and energetics [6].” Or something similar. The original sentence is too long and should be divided into 2 sentences. Please add a comma after „blood pressure” in line 68. Not „reduce” but „reduces” in line 69. Please add a comma before „decreasing water” in line 72. This sentence has grammar issuses: „Despite the strong long-term effects, exercise training is often unobserved and the its applicability on a large scale remains a major limitation.” Lines 73-74. Please change it to: „Despite the strong long-term effects, exercise training is often unobserved, and its applicability on a large scale remains a major limitation.” Lines 83-87. These sentences have grammar issuses: „Therefore the present study aims to evaluate the extended effects of a six-month AMPA program on metabolic syndrome patients and quantify its overall efficacy independently from gender, dietary habits and medications. Changing of anthropometric, cardiopulmonary and CV risk factor are considered.” Please change them to: „Therefore, the present study aims to evaluate the extended effects of a six-month AMPA program on metabolic syndrome patients and quantify its overall efficacy independently from gender, dietary habits, and medications. Changing anthropometric, cardiopulmonary, and CV risk factors are considered.”
2. Materials and Methods
2.1. Subjects The inclusion and exclusion criteria should be written coherently in a paragraph, bullet points should be avoided.
230 subjects were enrolled. This section must also say how many men and women were qualified for the study and how many met the criteria. This information cannot be found in the Results, although it can be briefly mentioned in the Results. Such information is found in the Methods section.
What is the ethicity of the participants?
3
2.2. Experimental Design
The following is italicized and it should not be: „This study utilized a prospective observational design, aiming to assess the effects of the AMPA program on various health outcomes. The primary objectives were to evaluate the impact of the exercise program on:” Lines 112-114.
Lines 112-118 should be one paragraph without bullet points.
There are phrases in bold. The should not be in bold: „This study utilized a”, aiming to assess the effects of the AMPA program on various health outcomes. The primary objectives were to evaluate the impact of the exercise program on”
2.3. Training Protocol
Please remove „ ,combining” and replace it with „that combined” in lines 131/132.
There are phrases in bold. The should not be in bold: „Aerobic Training:”in 135, „Resistance Training:” 144, „Each session included:” 152, „Warm-up:” 153, „Cool-down:”155.
Rewrite, please-lines 135-153. It should be in a paragraph with no bullet points. General Methods comments: Telemetry was mentioned in the Abstract. Where are details on how adherence to the individualized exercise plan was ensured or recorded?
3. Results
Please remove single phrases: „Correlations:” line 219, „Gender Differences” line 226, „Linear Regression Models” line 240, „Key findings from the models include the following:” line 246, „LDL cholesterol: „ 247, „Triglycerides:” 250, „HDL cholesterol:” 254. Results should be written without bullet points. If the Authors want to keep these phrases, I suggest creating subheadings.
Discussion
The study sample may lack diversity in terms of baseline fitness levels, which could limit the generalizability of the findings. This is a limitation. The following should not be in bold: „ In this study, we evaluated the effects of adapted and monitored exercise in 71 MS patients, following ACSM’s guidelines [6]. Our results show that Adapted Personalized Motor Activity (AMPA) [18] positively impacts MS and cardiovascular risk factors. Adherence to the program was high (77.4%), with less than one-fourth of participants unable to complete the six-month exercise program. Other studies reported higher drop-out rates”. Lines 263-268.
4
5. Conclusions
Limitations should be written at the end of the Discussion. Also- The sample size of 71 participants may limit the generalizability of the results. Next, the lack of a control group prevents definitive attribution of health outcome changes to the AMPA program. In the absence of a control group, it is challenging to exclude other variables that may have impacted the results. Next, The intervention's six-month duration and lack of long-term follow-up assessments hinder the evaluation of how lasting the observed health improvements are. Next, the decision not to provide dietary recommendations may limit the interpretation of changes in lipid profiles/ glucose levels. General manuscript comments: Why are reference numbers in red? Please write a point-by-point response.

Comments on the Quality of English Language

There are minor issues with commas and sentence structure, which I helped with. Please revise.

Author Response

1 Title The title of your manuscript should be concise, specific, and relevant. It should identify if the study reports (human or animal) trial data or is a systematic review, meta-analysis, or replication study. https://www.mdpi.com/journal/nutrients/instructions

We have revised the title to better reflect the study’s scope and methodology. The new title, "Tailored Exercise Intervention in Metabolic Syndrome: Cardiometabolic Improvements Beyond Weight Loss and Diet – A Prospective Observational Study," clearly specifies the study design and highlights the key finding that exercise benefits extend beyond weight loss. We believe this revised title provides a more precise and informative representation of our research.

Abstract The Abstract provides a sentence to place the question addressed in a broader context and highlight the purpose of the study. The Abstract explicitly states the study's main objective: to evaluate the effectiveness of a 6-month Adapted Personalized Motor Activity program in improving health outcomes in individuals with metabolic syndrome. A full stop is missing at the end of a sentence in line 35. Please write „The training” instead of „Training” in line 21.

We have made the following revisions:

  • A full stop has been added at the end of the sentence in line 35.

  • "Training" has been changed to "The training" in line 21, as per your suggestion.

These changes have been incorporated to improve the clarity and accuracy of the Abstract.

1. Introduction The Introduction effectively describes the rationale for the work in the context of existing knowledge. The Introduction provides an explicit statement of the objective by stating that the present study aims to evaluate the extended effects of a six-month Adapted Personalized Motor Activity (AMPA) program on patients with metabolic syndrome. It also specifies the goal of quantifying the program's overall efficacy independently of gender, dietary habits, and medications. The Introduction effectively defines metabolic syndrome and its key components, establishing its relevance as a public health issue. Citing global prevalence rates and the association of cardiovascular diseases with mortality is good. However, more specific studies might be needed to improve the reliability of the effectiveness of exercise compared to medications.

We appreciate your comment. To the best of our knowledge, no double-blind randomized controlled trials have directly compared the efficacy of pharmacological treatments versus structured exercise interventions in improving metabolic risk factors in individuals with metabolic syndrome. Furthermore, our objective is not to position these two strategies as mutually exclusive, but rather to underscore the importance of an integrated approach, combining pharmacotherapy and lifestyle modifications, to achieve metabolic targets more effectively and sustainably. Additionally, we have incorporated relevant literature on the effects of pharmacological treatments in the discussion to provide a broader perspective on this issue.

One of the keywords is „personalized exercise". This term does not exist in the Introduction.

Please delete „the” before „management” in line 43. Insert a comma before „and central obesity” in line 41. Lines 51-54: The following does not read well: „This means that improving VO2max through regular endurance exercise can play a critical role in enhancing cardiovascular fitness, reducing the risk of disease progression, and ultimately increasing longevity in both healthy sedentary and MS populations [7].” Maybe the Authors want to use the following instead: „This means that improving VO2max through regular endurance exercise can enhance cardiovascular fitness, reduce the risk of disease progression, and ultimately increase longevity in healthy sedentary and MS populations [7].”

We have made the suggested changes. Specifically, we have removed "the" before "management" in line 43, inserted a comma before "and central obesity" in line 41, and revised the sentence in lines 51-54 for better readability as follows: "This means that improving VO2max through regular endurance exercise can enhance cardiovascular fitness, reduce the risk of disease progression, and ultimately increase longevity in healthy sedentary and MS populations [7].

2. The following does not read well: „Endurance training helps to improve body composition by boosting daily energy expenditure and reducing visceral fat [4,6,8]; resistance training prevents muscle loss during weight reduction, increase fat-free mass, as well as maximal strength, muscular efficiency and energetics [6].” Lines 54-57. Maybe the Authors want to use the following instead: „Endurance training enhances body composition by increasing daily energy expenditure and decreasing visceral fat [4,6,8]. Meanwhile, resistance training helps prevent muscle loss during weight reduction while increasing fat-free mass, maximal strength, muscular efficiency, and energetics [6].” Or something similar. The original sentence is too long and should be divided into 2 sentences.

Please add a comma after „blood pressure” in line 68.

Not „reduce” but „reduces” in line 69.

Please add a comma before „decreasing water” in line 72.

This sentence has grammar issuses: „Despite the strong long-term effects, exercise training is often unobserved and the its applicability on a large scale remains a major limitation.” Lines 73-74. Please change it to: „Despite the strong long-term effects, exercise training is often unobserved, and its applicability on a large scale remains a major limitation.” Lines 83-87.

These sentences have grammar issuses: „Therefore the present study aims to evaluate the extended effects of a six-month AMPA program on metabolic syndrome patients and quantify its overall efficacy independently from gender, dietary habits and medications. Changing of anthropometric, cardiopulmonary and CV risk factor are considered.” Please change them to: „Therefore, the present study aims to evaluate the extended effects of a six-month AMPA program on metabolic syndrome patients and quantify its overall efficacy independently from gender, dietary habits, and medications. Changing anthropometric, cardiopulmonary, and CV risk factors are considered.”

We have made the following revisions as per your suggestions:

  • The sentence in lines 54-57 has been split and revised for clarity:
    "Endurance training enhances body composition by increasing daily energy expenditure and decreasing visceral fat [4,6,8]. Meanwhile, resistance training helps prevent muscle loss during weight reduction while increasing fat-free mass, maximal strength, muscular efficiency, and energetics [6]."

  • A comma has been added after "blood pressure" in line 68.

  • The verb "reduce" in line 69 has been corrected to "reduces."

  • A comma has been added before "decreasing water" in line 72.

  • The sentence in lines 73-74 has been revised for grammatical accuracy:
    "Despite the strong long-term effects, exercise training is often unobserved, and its applicability on a large scale remains a major limitation."

  • The sentences in lines 83-87 have been corrected for clarity and grammar:
    "Therefore, the present study aims to evaluate the extended effects of a six-month AMPA program on metabolic syndrome patients and quantify its overall efficacy independently from gender, dietary habits, and medications. Changing anthropometric, cardiopulmonary, and CV risk factors are considered."

2. Materials and Methods
2.1. Subjects The inclusion and exclusion criteria should be written coherently in a paragraph, bullet points should be avoided.
230 subjects were enrolled. This section must also say how many men and women were qualified for the study and how many met the criteria. This information cannot be found in the Results, although it can be briefly mentioned in the Results. Such information is found in the Methods section.
What is the ethicity of the participants?

In response to your suggestion, we have revised the Methods section to explicitly state that all participants were of Caucasian ethnicity. Additionally, we have included the specific numbers of men and women who were qualified for the study and met the inclusion/exclusion criteria.

2.2. Experimental Design
The following is italicized and it should not be: „This study utilized a prospective observational design, aiming to assess the effects of the AMPA program on various health outcomes. The primary objectives were to evaluate the impact of the exercise program on:” Lines 112-114.
Lines 112-118 should be one paragraph without bullet points.
There are phrases in bold. The should not be in bold: „This study utilized a”, aiming to assess the effects of the AMPA program on various health outcomes. The primary objectives were to evaluate the impact of the exercise program on”

We have revised the manuscript as follows:

  • The sentence previously italicized has been corrected and formatted properly.

  • The bullet points in lines 112-118 have been removed, and the information is now presented in a single paragraph.

  • We have also removed the bold formatting from the relevant phrases, including "This study utilized a," and the primary objectives.

2.3. Training Protocol
Please remove „ ,combining” and replace it with „that combined” in lines 131/132.
There are phrases in bold. The should not be in bold: „Aerobic Training:”in 135, „Resistance Training:” 144, „Each session included:” 152, „Warm-up:” 153, „Cool-down:”155.
Rewrite, please-lines 135-153. It should be in a paragraph with no bullet points.

General Methods comments: Telemetry was mentioned in the Abstract. Where are details on how adherence to the individualized exercise plan was ensured or recorded?

In response to your suggestion:

  • We have replaced ",combining" with "that combined" in lines 131-132 as recommended.

  • We have removed the bold formatting from the phrases "Aerobic Training:", "Resistance Training:", "Each session included:", "Warm-up:", and "Cool-down:".

  • The content from lines 135-153 has been rewritten in paragraph format, without bullet points, as requested.

Additionally, regarding your comment on adherence to the individualized exercise plan:

  • We have now added a section detailing how adherence was ensured and recorded during the study.

3. Results
Please remove single phrases: „Correlations:” line 219, „Gender Differences” line 226, „Linear Regression Models” line 240, „Key findings from the models include the following:” line 246, „LDL cholesterol: „ 247, „Triglycerides:” 250, „HDL cholesterol:” 254. Results should be written without bullet points. If the Authors want to keep these phrases, I suggest creating subheadings.

We have made the following changes:

  • We have removed the single phrases "Correlations:" (line 219), "Gender Differences" (line 226), "Linear Regression Models" (line 240), "Key findings from the models include the following:" (line 246), "LDL cholesterol:" (line 247), "Triglycerides:" (line 250), and "HDL cholesterol:" (line 254).

  • As per your suggestion, we have now created subheadings to organize the results, ensuring that they are written in a clear, structured manner without bullet points.

4. Discussion
The study sample may lack diversity in terms of baseline fitness levels, which could limit the generalizability of the findings. This is a limitation.

The following should not be in bold: „ In this study, we evaluated the effects of adapted and monitored exercise in 71 MS patients, following ACSM’s guidelines [6]. Our results show that Adapted Personalized Motor Activity (AMPA) [18] positively impacts MS and cardiovascular risk factors. Adherence to the program was high (77.4%), with less than one-fourth of participants unable to complete the six-month exercise program. Other studies reported higher drop-out rates”. Lines 263-268.

In response, we have made the following adjustments:

  • We have acknowledged the lack of diversity in terms of baseline fitness levels as a limitation of the study, as you suggested.

  • We have removed the bold formatting from the following section: "In this study, we evaluated the effects of adapted and monitored exercise in 71 MS patients, following ACSM’s guidelines [6]. Our results show that Adapted Personalized Motor Activity (AMPA) [18] positively impacts MS and cardiovascular risk factors. Adherence to the program was high (77.4%), with less than one-fourth of participants unable to complete the six-month exercise program. Other studies reported higher drop-out rates."

5. Conclusions
Limitations should be written at the end of the Discussion. Also- The sample size of 71 participants may limit the generalizability of the results. Next, the lack of a control group prevents definitive attribution of health outcome changes to the AMPA program. In the absence of a control group, it is challenging to exclude other variables that may have impacted the results. Next, The intervention's six-month duration and lack of long-term follow-up assessments hinder the evaluation of how lasting the observed health improvements are. Next, the decision not to provide dietary recommendations may limit the interpretation of changes in lipid profiles/ glucose levels.

General manuscript comments: Why are reference numbers in red? Please write a point-by-point response.

We have made the following revisions:

  • We have moved the limitations to the end of the Discussion section, as per your suggestion.

  • We have included the following limitations:

    • The sample size of 71 participants may limit the generalizability of the results.

    • The lack of a control group prevents definitive attribution of health outcome changes to the AMPA program. In the absence of a control group, it is challenging to exclude other variables that may have impacted the results.

    • The intervention’s six-month duration and the lack of long-term follow-up assessments hinder the evaluation of how lasting the observed health improvements are.

    • The decision not to provide dietary recommendations may limit the interpretation of changes in lipid profiles and glucose levels.

Regarding the red reference numbers, this issue was due to formatting errors during the document editing process. We have corrected this, and the reference numbers should now appear in the correct format.

Round 2

Reviewer 1 Report

Comments and Suggestions for Authors

The authors have made improvements to the article and taken into account the reviewers' recommendations, so I recommend publication.